# Inclusive Urban Mining: An Opportunity for Engineering Education

Sofia L. Schlezak *,† and Jaime E. Styer *,†

Humanitarian Engineering and Science MS Program, Department of Engineering, Design, and Society, Colorado School of Mines, 1318 Maple St. Bld #4, Golden, CO 80401, USA
* Correspondence: schlezaksofia@gmail.com or slschlezak@mines.edu (S.L.S.);
  jaime.styer22@gmail.com or jstyer@mines.edu (J.E.S.)
† These authors contributed equally to this work.

**Abstract:** With the understanding that the mining industry is an important and necessary part of the production chain, we argue that the future of mining must be sustainable and responsible when responding to the increasing material demands of the current and next generations. In this paper, we illustrate how concepts, such as inclusiveness and the circular economy, can come together in new forms of mining—what we call *inclusive urban mining*—that could be beneficial for not only the mining industry, but for the environmental and social justice efforts as well. Based on case studies in the construction and demolition waste and WEEE (or e-waste) sectors in Colombia and Argentina, we demonstrate that inclusive urban mining could present an opportunity to benefit society across multiple echelons, including empowering vulnerable communities and decreasing environmental degradation associated with extractive mining and improper waste management. Then, recognizing that most engineering curricula in this field do not include urban mining, especially from a community-based perspective, we show examples of the integration of this form of mining in engineering education in first-, third- and fourth-year design courses. We conclude by providing recommendations on how to make inclusive urban mining visible and relevant to engineering education.

**Keywords:** urban mining; circular economy; sustainable development; engineering education; humanitarian engineering; community development

## 1. Introduction

The first reference to urban mining is claimed to be in *The Economy of Cities* by the Urban Theorist Jane Jacobs in 1969 [1]. In her piece, the author described future cities as mines with huge, rich, and diverse raw materials [1,2]. However, the origins of this concept are still under discussion [3], and differences in definitions arise based on the contrasting ideologies and priorities of stakeholders yielding the term.

In general, an "urban mine" is understood as the urban accumulation of anthropogenic materials aboveground [2,4], and "urban mining" could be interpreted as the activity that converts "wastes" into resources [5]. Aldebei et al. [2] understand urban mining as a metaphor for describing the same activities of prospecting, exploration, development, and exploitation as traditional mining. For the purpose of this article, we will utilize the following definition proposed by Cossu and Williams, "Urban Mining extends landfill mining to the process of reclaiming compounds and elements from any kind of anthropogenic stocks, including buildings, infrastructure, industries, products (in and out of use), environmental media receiving anthropogenic emissions, etc." [6] (p. 1).

The term "urban mining" is assumed to be applicable to many kinds of waste [5]. However, this work is based on two specific streams, namely construction and demolition waste (C&DW) and waste of electrical and electronic equipment (widely known as WEEE

or e-waste). These are two of the most relevant anthropogenic sources in terms of quantity and economic incentive [7]. E-waste mainly motivates research and practice because of its high concentration of rare earth minerals, and buildings and infrastructure waste are the largest anthropogenic stock worldwide. In other words, C&DW is "the largest urban mine" [2] (p. 6).

As a direct consequence of the population growth, urbanization, and excessive consumption that characterize the last century, the exploitation of natural resources and the generation of waste have increased radically [5]. Under these circumstances, the concept of urban mining of anthropogenic wastes has been introduced for almost four decades as an alternative to the conventional way of extracting raw materials, which is particularly important to decrease their depletion and lower the mining footprint [8]. For example, managing electrical and electronic equipment under the circular economy approach can reduce the use of raw materials to produce new devices by up to 80% [7].

Urban mining also serves as an approach to sustainable waste management in cities and can be a source of new job opportunities for young people and/or immigrants [7]. As an anthropologist studying informal e-waste management in Tanzania observed, "( . . . ) recycling offers a skilled vocation, with a sense of stepped progression, secure revenue and entrance into a social support network that sustains and enhances local lives" [9] (p. 7). Therefore, urban mining also has the potential to become inclusive by contributing to the production of goods and offering services while simultaneously pursuing social objectives to enhance the quality of life of vulnerable communities.

Although it is promising as an economic, social, and environmental activity, urban mining still has its limitations. The recent efforts in legislation to promote urban mining that have been implemented in Europe and other regions, for example, the WEEE Directive (2012/19/EU) and its recent amendments that have become international models for e-waste management [10–12], are not enough to deal with the 82% of electrical and electronic equipment that is not treated in a sound environmental manner [7,12] and the 35% of C&DW that still ends up in landfills globally [13]. The causes of these figures include low recovery efficiency rates as a consequence of inefficient product design, lack of development and effective implementation of regulations and certifications to promote the use of reused materials, negative perceptions about second-hand materials and products, lack of awareness about the benefits of urban mining and the impacts of e-waste and C&DW, space scarcity in urban centers to store materials, high costs of best-quality recycling processes that make it difficult to afford for small and medium-sized enterprises (SMEs), and high competitiveness of landfilling associated with immature local markets and poor economic incentives for the circular economy [7,14].

In Latin America, despite advances in this field, much work still needs to be undertaken to improve the low e-waste recovery rate below 2% [12] and C&DW recovery rate below 10% [15]. As a result, a significant part of the waste with economic potential is abandoned in open spaces [2] or exported, resulting in lower efficiency of the waste management systems [16]. Furthermore, this situation limits green job opportunities in the region, especially for informal waste pickers and recyclers, who currently play a key role in the circular economy [17,18].

In the last decade, many international organizations have focused on this issue, and financial resources have been allocated in this field, e.g., the 2018–2022 UNIDO-GEF PREAL project for the Environmentally Sound Management of POPs in Waste of Electronic or Electrical Equipment [19]. However, projects and research are, in general, led by stakeholders interested in industrial ecology, waste management, environmental health, and the circular economy rather than academics and researchers from the mining sector with an interest in the potential of urban mining as an alternative economic activity of material extraction and social empowerment [18].

While the traditional training of mining, metallurgical, and materials engineers might not focus on urban mining [18], we agree that "shortly, our society is undergoing an accelerating transition from virgin mining of linear economy to urban mining of circular

economy" [20] (p. 104), and we suggest these groups should be part of this transition. Johansson et al. [18] claim that, as was the case for traditional and deep-sea mining, the development of technology could make urban mining attractive as an economic activity. They also highlight that even if the nature of mining changes, as with any mining activity, engineers and researchers specialized in materials composition, collection, extraction, separation, and recovery are crucial to overcoming current technological challenges for implementation.

In light of the need for further research and initiatives, there is a potential for science and engineering education to contribute to these global challenges. Some scholars reported a constant declining interest in mining studies worldwide [21] and proposed a focus on sustainable development to generate new competencies and subjects and promote innovative solutions and technologies by emphasizing environmental and social aspects [21]. Literature has reported the positive impacts of incorporating non-traditional mining areas into traditional engineering programs, for example, the case of incorporating artisanal and small-scale gold mining (ASGM) into the curriculum of an engineering college in the US [22,23], an approach that was also recommended by organizations, such as USAID and UNITAR as a crucial step towards formalization of the activity [24,25]. In this context, urban mining could also be proposed as an alternative to attract more students into the mining sector.

## 2. Objectives

Understanding that the mining industry is an important and necessary part of the production chain that should be aligned with international environmental agreements and goals, e.g., the United Nations 2030 Agenda and the Sustainable Development Goals, the future of mining must be sustainable and responsible when responding to the increasing material demands of the current and next generations. With this in mind, in this paper, we illustrate how concepts, such as inclusiveness and the circular economy, can come together in new forms of mining—what we call inclusive urban mining—that could be beneficial not only in mining engineering curricula and the mining industry but also for environmental and social justice efforts aimed at empowering vulnerable groups in regions, such as Latin America.

Population movement from rural areas to urban centers has created increasing demands for employment, often for individuals with low levels of education and literacy [26–28]. In this context, cities play a role in offering stable, secure, formal, economically sufficient, and dignified green jobs, including in the waste management sector [7]. Taking this into account, we begin by demonstrating how, if recognized, inclusive urban mining could present an opportunity to benefit society across multiple echelons, including empowering vulnerable communities (Section 5.1) and decreasing environmental degradation associated with extractive mining and improper waste management (Section 5.2). To do so, we use our research experiences in the C&DW and e-waste sectors in Colombia and Argentina to show how present and future urban miners are or can be empowered to build livelihoods out of treating what is traditionally seen as waste and how their path could be extended to prospective miners. Then, recognizing that most engineering curricula in this field (e.g., mining, environmental, and materials engineering) do not include urban mining, especially from a community-based perspective (Section 6.1), we show examples of the integration of this form of mining in engineering education in first-, third- and fourth-year design courses (Section 6.2). Finally, we conclude by providing recommendations for how to make inclusive urban mining visible and relevant to engineering education in different institutional contexts.

Given the few works found in the literature that introduce these ideas and perspectives, we challenge the status quo by proposing a conversation on a new paradigm that completely changes how we understand and treat material stocks. In this regard, we understand that the novelty of this study should be highlighted.

## 3. Description of Study Sites

### 3.1. C&WD in Colombia

In the past decade, there has been a rise in research and literature on C&DW management issues [29], and several countries, such as Germany, Spain, and Belgium are adopting strategies to treat and handle this type of waste [30]. However, Latin America lags in this area, and some countries, such as Colombia, despite generating vast amounts of C&DW, have not made noteworthy progress in managing it [31]. An estimated 35% of C&DW is disposed of in landfills without further treatment [32]. In their article, Colorado et al. attempt to obtain the first quantified values of C&DW in Colombia [13]. However, information on the management of C&DW in Colombia is very scarce, and Colorado et al. concluded that no reliable data depicting the amount of C&DW generated annually in Colombia exists. Similarly, most countries in Latin America do not collect data on the generation and quantification of C&DW [13].

Nevertheless, within a thesis project from 2003, author García Botero detailed C&DW in Bogotá, Colombia and examined if the sustainable development needs of the city are being met [33]. He concluded that approximately 99% of C&DW in the context of Bogotá is "useable", and a majority of this C&DW is made up of concrete, asphalt, brick, blocks, sand, gravel, earth, and mud. Furthermore, Méndez-Fajardo's article argues that recycling and reusing C&DW can produce significant positive impacts for citizens; however, these potential values are often overlooked [34]. These positive impacts can be seen at multiple echelons, including the environmental, social, cultural, economic, and even political level.

To study how to promote and support inclusive C&DW management in Colombia for the empowerment of low-income communities, we worked with Community A, a small, low-income community located just outside Girardot in the department of Cundinamarca. Girardot is a popular vacation spot and houses many recreational activities due to its warmer, tropical climate and proximity to Bogotá, the capital of Colombia which is home to over seven million people.

Unfortunately, not much is officially known about Community A. Based on our estimates, about 200 families live in the community, many of which do not have access to sewage systems. This site was considered relevant for our study because of their desire to take part in the project as well as the occupational profile of the inhabitants. While many of the men in the community work in construction practices, most of the women work in the informal sector (selling products, such as soda, avocados, and arepas from their homes or on carts either in the city or on the surrounding roads) or are unemployed. Furthermore, despite their occupational status, all the women in Community A are caretakers in their homes as well, for their children, parents, pets, and households. The local knowledge and the gender-related disparity in terms of job opportunities made Community A an interesting sample to study inclusive urban mining. The members in the community we spoke with wanted to learn how to extract value from C&DW and how they could make it profitable for themselves and their families. Thus, the goals of this project were refined with the guidance of the community, to increase education about C&DW and find a way to make this effort profitable.

### 3.2. E-Waste in Argentina

Despite the fact that Argentina had the highest generation of e-waste in 2019 (328 kt) out of the 13 countries studied by Wagner et al. [35], the management of this waste stream in Argentina is considered to be at a nascent stage, and little is known about it. Recent reports developed by national authorities confirmed a data gap [36], and the lack of regulations reveal that electronic waste management is a pending issue in the country. However, there are communities whose income depends on these materials. Some sources estimate that in 2017, nationally there were 600 people working in the informal e-waste recycling sector, and the number grew to around 2000 workers in 2019 [37]. A different source indicated that there were 2800 workers in 2019 in only 14 municipalities in the province of Buenos Aires [37].

The province of Buenos Aires and the City of Buenos Aires were selected as study sites in Argentina because they agglomerate the largest population and contain the enterprises and cooperatives that process the highest amount of e-waste [35,37]. In order to address the topic of inclusive urban mining, four cooperatives were studied. Cooperatives are "autonomous associations of persons united voluntarily to meet their common economic, social and cultural needs and aspirations through a jointly owned and democratically controlled enterprise" [38]. Three of the cooperatives under study are exclusively dedicated to e-waste, and one is a cooperative dedicated to solid waste but brings together workers who individually recover e-waste materials. Additionally, we included one university extension program currently offering e-waste management services in the province. The names and specific details of these facilities are protected, so they cannot be easily recognized.

- Facility A (cooperative): It started in 2018 as a solid waste cooperative, and since 2022, its members have decided to explore e-waste processing as an additional source of income. The e-waste sector now has five workers and one coordinator. They are in the process of formalizing their activity in relation to e-waste management.
- Facility B (cooperative): It is a solid waste cooperative that started almost ten years ago. They have more than 150 members, and almost half of them individually recover e-waste material from the streets. This cooperative is interested in e-waste, but its members do not yet have experience with this waste stream.
- Facility C (cooperative): With almost 20 workers, this cooperative is one of the province's most advanced small social businesses. They have already obtained legal permission to manage e-waste, their main activity.
- Facility D (cooperative): It has over 20 years of experience in the business and employs more than 25 workers. The cooperative is recognized as a formal e-waste operator and treats almost 1,400 tons of waste per year.
- Facility E (University extension program): It began as an academic extension program and now is one of the few e-waste operators in the province. Since 2009, they have trained 1168 students and treated 217 tons of e-waste.

## 4. Methods

### 4.1. Methodological Framework and Ethical Considerations

As the local knowledge of communities is crucial for developing sustainable and just solutions [39], our research methods were participatory and we took a qualitative approach to understand the neglected knowledge, expertise, and values of the communities we worked with, as well as the complex systems that shape their lives. Qualitative methodologies, such as semi-structured interviews, participant observation, focus groups as well as workshops, were utilized to build this understanding. Figure 1 summarizes the logical items of this study.

Both projects were approved by the Colorado School of Mines Human Subjects Research Team and exempted from the Institutional Review Board (IRB) process requirements. The research in Colombia was developed in partnership with a Colombian university, Corporación Universitaria Minuto de Dios or Uniminuto. The groups specifically working on this project were the Parque Científico de Innovación Social (PCIS)/the Social Innovation Science Park, a research group led by Civil Engineering and Occupational Health and Safety Professors with social justice aims entitled Ingeniero a tu Barrio, the international studies group in Uniminuto-Girardot, as well as communication specialists including Professor Martha Liliana Herrera Gutiérrez, who was responsible for translation, facilitation, and communication. Together we worked directly with a low-income community in Colombia, Community A, to study how recycling C&DW, specifically concrete, could empower them. Approval for this research was also obtained from local Colombian authorities, including Uniminuto's PCIS as well as their research ethics committee. Additionally, the research in Colombia adheres to Uniminuto's Social Innovation Route Framework [40], a five-phase community engagement framework developed by PCIS.

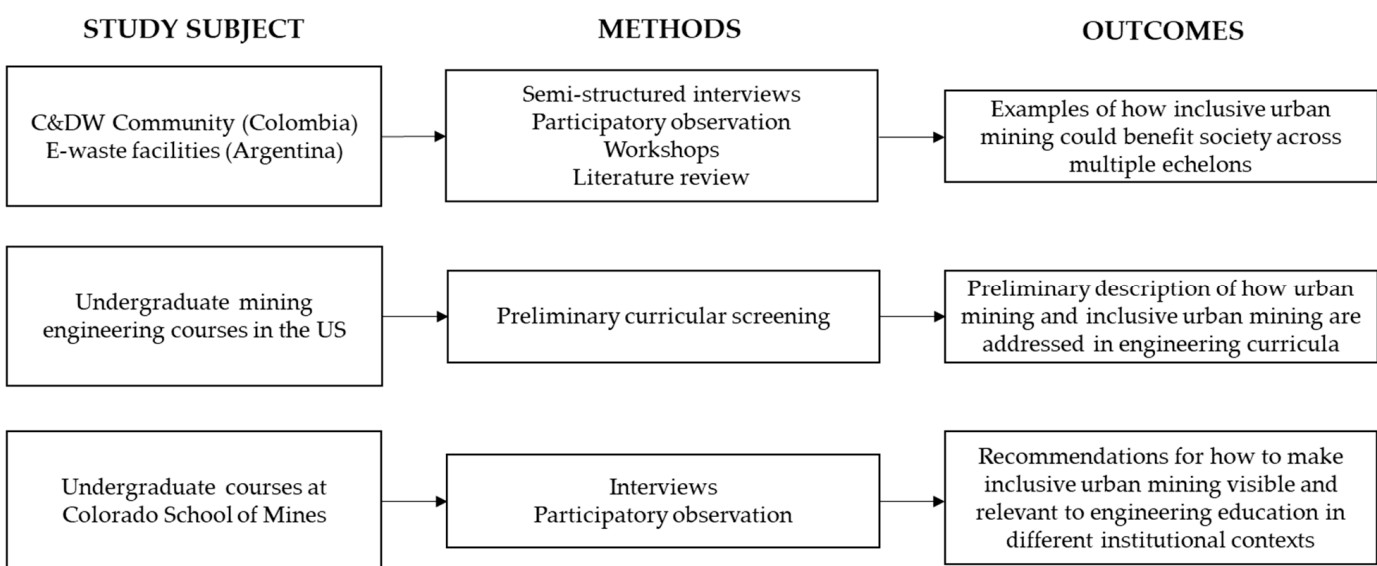

**Figure 1.** Schematic representation of the logical items of this study.

*4.2. Semi-Structured Interviews*

Throughout the six weeks of fieldwork on C&DW completed in Colombia during June and July 2022, we interviewed 17 women and 12 men from differing backgrounds, including low-income community members, community leaders, engineers, academics, students, waste management experts, and government officials. During each interview, translation services were provided by professors from Uniminuto. Within these interviews, we asked our interlocutors to describe their backgrounds, knowledge of concrete and C&DW, risks and barriers they believed could prevent recycling C&DW in low-income communities, and if any groups could be disproportionately affected by these risks. Additionally, as this project had specific social justice goals to contribute to women's empowerment in low-income communities, our focus was to speak mostly with women in the community to understand their interpretations of women's empowerment and define how they specifically wanted this project to empower them.

A series of exploratory interviews were conducted during June and July 2022 for the research on e-waste. A total of 15 government representatives, researchers, and members of e-waste cooperatives and programs were interviewed. The meetings, each lasting approximately one and a half hours, were held virtually. The interviews aimed to obtain preliminary information on the current situation of e-waste management in Argentina, with a particular focus on the province and the city of Buenos Aires. Participants were asked to describe their workplaces, e-waste-related practices and dynamics, the challenges the sector faces, and their opinions on past and current waste management strategies. Additional unrecorded interviews with e-waste workers and government officials were also conducted during the participatory observation visits described in the next section.

*4.3. Participatory Observation*

Upon beginning our fieldwork session in Colombia, the Uniminuto team immediately began facilitating meetings with Community A. As the Uniminuto team had already collaborated with this community in the past, many community members already knew most people on our team. However, building rapport with the community during these meetings was essential for the group members who had not previously worked with Community A. Thus, time was spent introducing ourselves and our goals and conversing with the community members. To understand the community's goals for this project, it was essential to understand the context of the community, their values, beliefs, journeys, destinations, language, knowledge, and more through participant observations. We spent a lot of time trying to understand the knowledge community members had about C&DW

and recycling. We learned the community was already utilizing C&DW in their homes and on shared roads but in unsustainable ways. While there is a Junta de Acción Comunal for the community (a legally protected organized civil structure made up of members of the community), there are also natural leadership structures and leaders as well. Despite this divide in power structures, both groups wanted to find a way to make recycling C&DW profitable and beneficial for the community. The understanding of these relationships contributed to the understanding of urban mining potential in the community.

In Buenos Aires, jointly with the representatives of two local government agencies, three visits to e-waste facility A and two to facility B were conducted in August 2022. In facility A, the researcher was accompanied by a team consisting of one toxicologist and three social workers, and in facility B an environmental professional led the visit. Three other e-waste facilities in the province of Buenos Aires were visited by the researcher. During these encounters, neither video nor audio recordings were made. Photographs of the workspace, machinery, devices, and waste were taken with the previous authorization of participants. The objective of this observation was to understand the different contexts of e-waste workers, their work dynamics, power distributions, needs, concerns, and desires. These interactions helped identify new actors and refine data on materials and equipment, collection and treatment practices, and the value chain characteristics.

*4.4. Workshops*

Through the preliminary analysis of the data gathered on C&DW and Community A during interviews and participant observations, we formulated a common theme, which was the desire to bring educational opportunities to the community and conduct workshops to give community members, especially women, skills to generate income. To follow our commitment to a community-centered research approach, we developed a participatory workshop with Community A that took place in March 2023. This workshop presented an opportunity for women and low-income community members in Colombia, specifically targeted towards Community A in this approach, to engage one another in the process of learning about recycling concrete from C&DW and develop a plan for how this can become actionable in their community. The workshop had five key sections: C&DW Composition and Values, Environmental Aspects, and Necessary Permits; C&D Recycling Processes and Technologies; Occupational Health, and Safety; Applications and Entrepreneurism; and Pathways Forward. We worked with community members through virtual communications and surveys to ensure that the included components were necessary and relevant. Additionally, to centralize local knowledge and build local capacity we invited Colombian subject matter experts to lead each of the key sections.

For e-waste communities, two workshops were held with each group of workers from facilities A and B. The main objective of the workshops was to analyze workers' perceptions regarding the chemical risks related to their activity, train them in basic concepts of risk prevention and management, and collect their opinions on a proposed intervention to prevent the open burning of cables. All the e-waste workers at facility A (five males) participated in the workshop with their coordinator (one male). At site B, since the cooperative is not formally working with e-waste, the associated urban recyclers with e-waste experience were invited to participate. In total, 37 (17 females, and 20 males) and their coordinator (1 male) participated in the workshop. The activities were conducted in two hours and included: (1) Initial general risk identification activity, (2) E-waste risk perceptions activity, (3) Discussion about a cable stripper prototype, and (4) Community mapping of burning sites, metal buyers, and collection points (only at site A). Audio recordings were taken with the prior permission of the participants.

*4.5. Research Extension with the Undergraduate Students of the Colorado School of Mines*

Following the teaching philosophy of the Colorado School of Mines (CSM) Humanitarian Engineering and Science program [41], three graduate–undergraduate research extension activities were carried out with CSM undergraduate students. First, the topic

"Empowering People: Extracting Value from Waste Through Urban Mining" was proposed as project motivation in the "Design I" course in Fall 2022, aimed at more than 600 first-year engineering students. Second, for a senior design course, specific sociotechnical challenges related to C&DW and e-waste were proposed to the students. Third, a new version of the "Engineering and Sustainable Community Development" course was delivered to 24 undergraduate students in Spring 2023, based on three e-waste technical challenges defined by a community of recyclers in Bogotá, Colombia.

*4.6. Preliminary Review of the Mining Engineering Curricula*

To describe the approach of urban mining and inclusive urban mining as topics in the engineering curricula, we conducted preliminary research on curricular databases from universities in the United States. We selected the top universities in mining engineering, as listed on the National Mining Association website [42], and then we examined each university's website and online course catalog individually, solely looking at their minimum requirements for obtaining a Mining Engineering Bachelor of Science in the 2022–2023 academic year. University, general education, and B.S. course requirements were not included. When examining each catalog, we searched each course description to determine if urban mining concepts and sociotechnical approaches were explicitly stated as learning goals in required courses. To search for urban mining concepts, we utilized the following search terms: "urban mining", "construction and demolition waste", "C&DW", "electronic waste", and "e-waste". We also searched for sociotechnical learning approaches by searching for the terms "community", "sociotechnical", "social", "societal", or "human". Acknowledging the limitations of this approach because of its subjectivity related to the lack of detailed information describing the material reviewed within the required courses as well as the research projects being conducted within these universities external to course curricula, we incorporated our findings as a preliminary set of data that could be further analyzed in future works.

**5. Inclusive Urban Mining in Latin America**

Currently, in most Latin American countries, the "resource recoverers" [43] or, as we refer to them in this paper, "urban miners", are not yet integrated into the regulatory framework. Their working conditions are often precarious, exposing them to hazardous chemicals, including heavy metals and halogenated compounds [44]. Although they provide a critical environmental service, waste workers have historically been stigmatized and excluded within society [43].

With growing interest, but still nominal in comparison with traditional mining, some countries in the region are facing the challenge of regulating the activity of urban mining, integrating informal workers, and promoting improvements in processes and technologies to increase productivity and promote sustainable local economic development [35,36].

To help overcome the challenges and barriers enumerated, we present below the current and potential benefits of urban mining for communities and the environment with a special focus on Latin America. We support our claims by providing evidence from our literature research and experiences with communities in Argentina and Colombia interested in treating waste not only as a source of income but also as a way of empowerment. We introduce inclusive urban mining as a concept that has its roots in inclusive solid waste management, an activity with a long history in these two countries [45,46].

*5.1. Why Should Urban Mining Be Inclusive? Some Examples of Its Social Benefits*

5.1.1. Women's Empowerment: Through Recycling C&DW

When asked about their role in their community, household, and workplace, many of the women in Community A in the Colombia research claimed to be a leader of some kind. They either defined that as having a position of power and knowing they could tell others what to do, or being a mentor or a friend to people when they needed something. Some also cited their age when asked about their role, stating that because they are older,

they are wiser and therefore better leaders. When asked about the problems women face in their community, problems with children were often cited, such as children being left alone or turning to illegal activities to make money. Another problem that often arose was unemployment, sometimes due to the lack of transportation or job opportunities. Finally, when asked about solutions to these problems and how the term women's empowerment was understood, people often brought up workshops that had been conducted in the community in the past. These workshops often focused on cosmetology, such as doing hair and nails, art practices, baking, or cooking. Many people mentioned education and the importance of learning, and some brought up making money and having the ability to secure and spend money for themselves or their family while in the confines of their own homes. When asked about women's empowerment, one leader in the community stated, "[Women's empowerment] is the idea that women can work on their own with their own capacities". She also discussed the importance of bringing opportunities to the community and doing workshops to give women tools to find jobs.

As illustrated in the semi-structured interviews conducted with women in Community A, in this context, it was found that urban mining can best empower them helping them generate more income and advance their education to gain additional skill sets. While financial and economic decision-making power is a common dimension of women's empowerment, the details of how exactly this pursuit could be more beneficial and empowering to women in Community A were developed through a dialogue and an understanding of the community context. For example, the women demonstrated the need for the time and capacity to care for their families alongside these pursuits, thus making this a homebound endeavor.

While urban mining shows promise to contribute to empowerment opportunities for multiple vulnerable groups, including women, it must be acknowledged that these contributions can be maximized through a contextual understanding of the complex systems that shape the lives of these groups. As such, there is a need for academic institutions, especially those related to engineering and design, to work alongside communities to understand how pursuits, such as urban mining, can empower them. Moreover, to ensure relevant empowerment to vulnerable groups, it is essential to take an interdisciplinary approach, as empowerment is contextually situated; thus, different ideas of empowerment exist within different contexts and are reflective of their own specific communities and cultures.

5.1.2. Social Transformation and Digital Inclusion in the E-Waste Sector

In our visits to e-waste facilities, we observed the pride of workers regarding their role as green actors in a context where circular economy policies and regulations, although necessary, are still pending. This role is one of their motivations when facing the many obstacles presented to them. To name a few, they have little bargaining power vis à vis buyers—mostly intermediaries—and limited access to information and technologies to maximize waste recovery. Even with all the challenges, these groups of workers, mostly born and raised in vulnerable conditions, go through a collective process of what they call "subsistence, resistance, and transformation". Some are young adults who have never kept a job for more than a couple of months, but in their cooperatives, they become resilient and learn not only specific knowledge relevant to their business, but also the general rules of the labor industry, such as complying with the schedule and attendance. Hence, as an interlocutor told us once, "We [the cooperative] not only recycle materials, but people". Therefore, it is not arbitrary that some cooperatives have included words, such as "dignity" or "justice" in their names. We see, then, that the feeling of belongingness that the activity generates in workers has the potential to contribute to the education on labor conduct, becoming a transforming process for specific groups, including young recyclers.

For workers in general, as in any other labor space, learning new skills and developing new knowledge are essential, and urban miners are not exempted from this process. They learn to repair and disassemble e-waste by gaining specific knowledge about electronics, IT, mechanics, and sometimes material composition and chemistry. However, particularly for this sector and especially in the Latin American context, learning these disciplines goes

beyond training workers in their roles. This learning process also means a step towards their insertion in an increasingly demanding digital society. This is an additional benefit of urban mining, illustrated by the case of a worker who, during a workshop, told us, "Since they [the cooperative] gave me a computer, I was able to use one for the first time in 60 years". This worker's access to technology, although based on the objective of training him on electronics repairing, ended up meaning his access to a digital world that is often hampered for people his age.

In urban mining cooperatives, we have observed how not only materials but also people are transformed. Understanding this opens the way to many study areas that are scarcely explored today by the academic community. We question what other social benefits do urban mining cooperatives bring? Could the social benefits be externalities that account for the comparison between urban and traditional mining? We wonder, in particular, for the e-waste management sector, how could actors dedicated to digital inclusion and actors dedicated to e-waste management interact? What impacts would a more inclusive digital society have on the use, disposal, and management of electrical and electronic devices? Could the circular economy based on e-waste become a mechanism for digital inclusion?

5.1.3. The Value of Local Knowledge for the Global Development of Urban Mining: Examples from the E-Waste Sector

Although many of the e-waste recoverers did not perceive themselves as producers of knowledge or technology, as a result of our visits and workshops, we learned that the knowledge of these workers is as important as any other certified by an academic degree. For example, some workers can quickly identify components and materials with high efficiency, and some apply craft and ingenious low-cost plastic identification methods (e.g., by their texture, smell, or color). Others have perfected techniques, such as manual disassembly or burning, to improve the quality of the metals they obtain, even in conditions that create major health problems for them and their communities because of their exposure to hazardous chemicals [44]. Likewise, some workers with more than a decade of experience in the sector have undertaken the important task of sharing their knowledge with less experienced peers, providing in-person training and written material. The information is exchanged between the workers themselves. They themselves are the referents of the activity and share their knowledge. A good example is the free and public guidance document "Cooperación y reciclado para un mundo sustentable" ("Cooperation and recycling for a sustainable world") edited by Salcedo et al. in 2019 [47].

In the literature, waste workers are usually pigeonholed into informality [48], and under a global gaze that proposes external strategies to deal with local problems. We believe that to avoid the traditional labels of "lacking" or "informal," and put an end to the historical marginalization of the recyclers, waste pickers, and waste workers in general, it is necessary to study their resilient learning, improvement, and knowledge-transfer processes. Johansson et al. [18] claim that "the informal sector can nevertheless teach us how to change our perception of technospheric stocks and view them not as a problem but as a resource" (p. 42). We thus wonder how their voices could be amplified so that larger audiences know how their inventiveness and persistence can overcome the barriers of the context in which they work and how these skills and knowledges can help alleviate the global challenge of e-waste, C&DW, and other relevant waste streams. We then ask what can Latin American urban miners contribute to the global conversation on waste management? How can local knowledge improve foreign technological processes? These proposals are not in opposition, but on the contrary, they seek to promote a synergistic interaction between the development of knowledge and technologies in Latin American countries and countries of other regions.

*5.2. Environmental Benefits of Urban Mining*

Present-day demand for material resources combined with concerns about the sustainability of extraction practices and the effects of waste have increased the interest of both practitioners and scholars in the concept of the circular economy. In this context, urban

mining is gaining momentum from various perspectives [49]. First, this practice rejects linear approaches to production, replacing the "end-of-life" stage of traditional waste management with reusing, recovering, and recycling processes throughout a product's life [50]. Second, many scholars agree that urban mining improves resource fulfillment by advancing the circular economy and minimizing environmental burdens [51]. Obtaining materials from discarded items can also contribute to climate change mitigation since metal recovery consumes less energy than the extraction of primary raw materials [51]. For instance, the energy needed for the manufacturing and transportation of building materials could be reduced by about 29% if these materials are recycled [7,52,53]. Third, urban mining provides a solution for uncontrolled waste management, which remains a significant global challenge [14] due to factors, such as the exposure of the environment and humans to hazardous substances and biological vectors. The accelerated growth of waste on a global scale results in valuable aboveground stocks in quantities that are often comparable to or exceed natural stocks [6]. For example, Grant et al. [3] indicate that "thirty smartphones contain as much gold as one ton of mine rock from a traditional gold mine" (p. 7). Thus, these resources have become attractive to those that acknowledge the gradual depletion of economically minable resources [49].

Environmental Benefits of Recycling C&DW in Colombia and E-Waste in Argentina

The construction industry is a main contributor to carbon dioxide emissions across the globe due to it containing many elements with high carbon footprints, such as cement and concrete production, transportation, and C&DW generation [54]. In 2020, the United Nations Environment Program (UNEP) stated that the buildings and construction sector accounted for 38% of the total global energy-related $CO_2$ emissions in 2019 [55]. The cement industry alone contributes to about 8% of the global $CO_2$ emissions [56]. Effectively managing C&DW is a critical component of preserving our environment, natural resources, economy, and society [57]. Despite this, C&DW mismanagement is a widespread issue.

Around the world, the problem of C&DW is worsening, thereby exacerbating environmental and social issues [58]. In Colombia, the expansion of the construction industry is aggravating these issues through the disposal of C&DW in an insufficient and unregulated manner and the increased illegal extraction of aggregate materials [59]. These increasing environmental and social issues are gaining national recognition in Colombia, as seen in Resolution 472, which outlines the management of C&DW in Colombia in light of the inadequate disposal and increased generation of C&DW in cities across the country, including Bogotá, Medellín, Santiago de Cali, Manizales, Cartagena, Pereira, Ibagué, Pasto, Barranquilla, Neiva, Valledupar and San Andrés [60] as well as other legislation released over the past couple years [61–63]. Recycling C&DW could contribute to reducing the inadequate and unregulated disposal of C&DW and decrease the illegal extraction of aggregate materials.

Regarding e-waste in Argentina, a national report estimates that 465,000 tons of this waste stream are generated per year [36] and only 4% is managed in an environmentally sound manner [35]. Roughly, following the methodology in Forti et al. [12], we calculated that this low percentage contributed to a net saving of 8 kt of $CO_2$, equivalent to emissions from the recycling of secondary raw materials substituted to virgin materials. If this percentage increases up to the goal of 30% under Target 3.2 of the ITU Connect 2030 Agenda [64], it might help save up to 60 kt of $CO_2$ equivalent emissions.

## 6. Inclusive Urban Mining in US Engineering Curricula

### 6.1. Preliminary Screening of Urban Mining Content in the US Engineering Programs

Table 1 summarizes the preliminary research findings on curricular databases from the universities in the United States with Mining Engineering Bachelor of Science programs.

**Table 1.** Preliminary review of Mining Engineering Curricula related to urban mining and sociotechnical learning approaches * in the US.

| | Mining Engineering University | | | | | | | | | | | | |
|---|---|---|---|---|---|---|---|---|---|---|---|---|---|
| **Number of Required Courses** | **1** | **2** | **3** | **4** | **5** | **6** | **7** | **8** | **9** | **10** | **11** | **12** | **13** |
| Urban Mining Concepts | 0 | 0 | 0 | 0 | 0 | 0 | 0 | 0 | 0 | 0 | 0 | 0 | 0 |
| Sociotechnical Concepts | 0 | 2 | 0 | 1 | 0 | 1 | 0 | 4 | 3 | 1 | 0 | 0 | 0 |
| **Number of Non- Required Courses** | **1** | **2** | **3** | **4** | **5** | **6** | **7** | **8** | **9** | **10** | **11** | **12** | **13** |
| Urban Mining Concepts | 0 | 0 | 0 | 0 | 0 | 0 | 0 | 0 | 0 | 0 | 0 | 0 | 0 |
| Sociotechnical Concepts | 1 | 3 | 3 | 2 | 0 | 2 | 0 | 4 | 3 | 1 | 0 | 0 | 1 |

* Search terms: "urban mining", "construction and demolition waste", "C&DW", "electronic waste", "e-waste", "community", "sociotechnical", "social", "societal", "human".

There are 13 universities in the United States with Mining Engineering programs. Within these programs, urban mining was not explicitly listed as a learning goal in any course descriptions found within the universities' websites or online course catalogs depicting the minimum course requirements for obtaining a Mining Engineering Bachelor of Science in the 2022–2023 academic year. The programs include courses with sociotechnical approaches, or at least discuss human-based concepts to a certain extent; however, these approaches were found more often within the non-required courses.

To reiterate, our findings should be viewed as a preliminary set of data that could be further analyzed in future works due to the limitations of this approach, including its subjectivity related to the lack of detailed information describing the material reviewed within the required courses in course descriptions found on university websites. Additionally, this screening did not include research projects being conducted within these universities external to course curricula.

It is not the intention of this work to only focus on the US curricula but to call on global engineering and technology academic institutions to involve current waste management challenges in their programs as motivators for technological innovation projects. Through the examples described below, we want to emphasize that urban mining could be a significant research subject and an excellent educational opportunity for organizations specializing in traditional mining, as these organizations could take advantage of their existing technical knowledge in material extraction and processing.

### 6.2. Approaches to Include Urban Mining in the US Engineering Curricula

6.2.1. Introducing Urban Mining to First-Year Engineering Design Courses

The faculty in the design 1 and 2 courses at CSM are diverse in terms of academic and professional backgrounds. They mostly have STEM majors, such as design, civil engineering, electrical engineering, and other traditional engineering disciplines. Some have experience in the industry, and some others are senior researchers. This diversity gives the students exposure to different industries and "real-world problem-solving and design experiences", as our interviewee claimed. In total, they teach 25 groups of 25 first-year engineering students by applying an ambitious but enriching approach involving problem formulation, design thinking, and stakeholder engagement.

The purpose of their teaching approach is to expose students to stakeholders and communities they did not know about and to make them reflect on how engineering projects might affect those communities. Its purpose is also to broaden students' perspectives in ways they might never find out on the news or social media. This approach is not exempt from resistance, either from the faculty or students. According to the instructor, many students tend to separate the technical and the social and only focus on the technical challenge because, in the end, "the majority is going to end up in traditional engineering jobs one day, and that is just the path they want". Some other students understand the complex issues that low-income communities are facing, but they just do not want to get involved. In this context, the instructors try to emphasize the importance of integrating knowledge. They

explain to students that engaging with stakeholders and considering the specific contexts, geographies, cultures, and expectations are key stages in the design process. "You cannot do technical in a vacuum", our interviewee claimed. As an example, he asks students, "Could you design a technical solution without considering government regulations"?

The efforts of going beyond the boundaries of traditional engineering are huge for the faculty, and despite some room for improvement in terms of genuine stakeholder engagement, those instructors with traditional engineering backgrounds are proud of what they are doing in this class. They are aware that even if it is an introductory course, they provide additional techniques that are not usually offered in the first year in other programs in the US.

At CSM, students receive a "call for proposal" (CFP) broad enough so they can elaborate on the problem after a series of research stages that can involve literature research and consultation with subject experts, potential users, and other stakeholders. In 2022, for the very first time, the CFP was developed in collaboration with two graduate students from the Humanitarian Engineering and Science program, who are the authors of this paper. The topic, "Empowering People: Extracting Value from Waste Through Urban Mining" was innovative since it introduced urban mining, life cycle, and waste management as motivations for design. The students received a brief description of the general situation of waste management systems in low-income communities and the specific challenges and opportunities in the study sites that we present in this paper.

The way the CFP was developed allowed students to set the boundaries of the problems by themselves, encouraging them to think creatively and "out of the box", to look at things outside their own immediate context, and to familiarize with the processes that happened after "the magic truck comes by and picks up the purple bin".

From a total of approximately 625 students, 110 students grouped in 25 teams achieved the 20% best-scored projects. Among the winning teams, the distribution of themes was Food/organic waste (5), E-waste (4), Plastic/packaging waste (4), Household effluents (2), Medical waste (2), Textile waste (2), and Others/Out of scope (6).

The CFP motivated students to reflect further down the line at the end of product life. For example, some students worked on recycling technologies to be applied locally. Others looked at extending the life cycle by redesigning products or tried to look for upcycling opportunities at the source. A number of students preferred to address the specific challenges related to the settings and contexts that we have presented. They usually choose their path according to what they are exposed to and tend to lean towards stakeholders that they already know. The groups interacted with recycling, electronics, processing companies, big warehouses, consumers, and professional users.

Although the experience was enriching for both students and professors, introducing a new way of mining provoked tensions in a school well-known for its mining tradition. Some mining professionals asked, "Why are they calling this mining"? and claimed, "This is not our definition of mining. This is not what mining engineering is". We wonder, then, what does it take for the traditional disciplines to extend their boundaries? In the end, changes in the curricula that in the past seemed far off, such as the inclusion of the social aspects in a first-year engineering design course, became a reality seven years ago. We wonder, what is urban mining lacking to be considered as a topic of relevance by the traditional mining sector? What are the differences? What are the convergence points?

6.2.2. Introducing Urban Mining in Elective Courses: The Case in an Engineering and Sustainable Community Development course

For the very first time, in Spring 2023, the course Engineering and Sustainable Community Development (ESCD) was taught in a project-based format, involving direct interaction with communities. This course gathered twenty-five third-year and graduate students to work collaboratively with a Colombian recycling association to improve three processes: e-waste plastic identification, copper separation from cables, and precious metals separation from circuit plastic boards. From the instructor's perspective, urban mining is not the initial

motivation of students that join this course. These students care about community-based projects in general, and even if they have education in specific disciplines (Environmental Engineering, Civil Engineering, Mechanical Engineering, Chemical Engineering, and Design Engineering), they are curious about the different ESCD opportunities for practice. Addressing a waste-related topic and how other communities interact with it stimulates students to think in ways they never have.

According to the instructor, urban mining could be framed as the future of mining. He thinks it has the potential to convene students, researchers, and industry professionals who are not usually involved in traditional mining (for example, electrical and civil engineers) that would be able to apply their knowledge and skills to transform anthropogenic waste stocks into valuable materials. In this sense, traditional mining institutions could see urban mining as a way to expand their curricula, staff, and areas of expertise. The expansion, however, will not be easy for those that have a traditional mining background, he said. It will require not only their willingness but the comprehension of new knowledge to deal with mines not located in the mountains but in the cities. Therefore, the challenge ahead will be to deal with the technical differences as well as the intricate relationship between material extraction and urban systems.

When we asked him how to make urban mining visible and relevant, the instructor did not hesitate to claim that extending graduate research into first and third-year design courses is an important grassroots step that could eventually position the topic as an institutional priority from the top-down. He explained that improving urban mining not only contributes to the circular economy but also provides employment opportunities for marginalized groups of people, such as those displaced by violence, poverty, or climate change. In this sense, applying a community-based approach in the engineering curricula gains more significance when urban mining is seen as an employment solution. He acknowledged that engineers could address urban mining from an industrial, automated, and large-scale perspective, but in doing so, they might be ignoring and neglecting the current labor problems that cities are facing and the minor waste streams that are managed in small neighborhoods. "All those stocks of waste are always going to exist, and all those people needing employment are always going to exist irrespective of the big machinery that you put in place". Hence, he emphasizes the need for more engineers and engineering students to be trained to co-work with marginalized communities with the aim of improving their processes, products, and labor conditions.

The instructor also pointed out some important parallels between artisanal small-scale gold mining (ASGM) and urban mining. Less than a decade ago, the Minamata Convention forced countries to focus on reducing and, when feasible, eliminating the use of mercury. ASGM became a major area of interest for many institutions, including well-known traditional mining schools and research centers in the Global North. This new area of interest opened opportunities for research and practice in fields, such as engineering and social sciences. The recurrent presence of informality and the way in which communities engage in these activities, sometimes ending up exposing them and their families to hazardous chemicals, are other points of commonality between ASGM and urban mining. Furthermore, ASGM and urban miners both "are for the most part invisible to mainstream society", the professor said. Taking this into account, we wonder if similar drivers, such as the global concern about scarce materials, including rare earth elements, metals, and minerals, might have the same result for urban mining. Will inclusive urban mining become a field of research and practice in the way that artisanal scale gold mining did, even with the tensions and resistance that it generates?

### 6.2.3. Introducing Urban Mining Projects in Third and Fourth-Year Project-Based Design Courses

In an effort to offer the upper-level engineering students opportunities to learn more about human-centered design and humanitarian engineering challenges, CSM offers a three-semester hour project-based design course targeted towards junior- and senior-level

students. Within the Fall 2022 semester, multiple graduate students from the Humanitarian Engineering and Science program—including the authors of this paper—were able to work with project teams in this course on specific real problems affecting real people. Overall, of the 23 students registered for this course in the Fall 2022 semester, 10 students worked on urban mining-related projects. One group of four students worked on a C&DW-related project, while two groups of three students were devoted to e-waste-related projects.

The faculty member responsible for facilitating the course in the Fall 2022 semester described urban mining as an opportunity, not only to "emphasize reclamation of precious materials in environmentally friendly ways that are also economically beneficial to disadvantaged populations" but to push back on the negative connotation associated with the term "mine" due to the often-harmful activities, practices, and ramifications of the industry. The professor believes urban mining is a way to "reclaim the word 'mine' for positive applications", and institutions responsible for the progression of the often damaging activities, practices, and ramifications of traditional mining processes, such as universities including CSM, should be at the forefront of developing "more environmentally friendly and socially equitable ways of mining and engineering", such as urban mining. The inclusion of urban mining in the curricula to atone for the negative externalities involved in traditional mining can also be an opportunity to enhance mining engineering education by facilitating understandings of concepts, such as life cycle analysis.

In addition, the professor stressed the importance of understanding the local context of projects, such as the cultural, socioeconomic, and environmental dimensions of the cities with which they work, and utilizing approaches from the social sciences and environmental sciences to develop solutions that are "most appropriate to their target population and do the least harm to the same population as well as their environment". He stated that this utilization and understanding was even more essential than technical foundations, such as the engineering mindset, to arrive at a point where the technical solutions were appropriate. Furthermore, to develop the best solutions possible, the professor argued that stakeholder engagement, particularly empathetic stakeholder engagement ("which is culturally sensitive, appropriate for local contexts, aware of potential unintended consequences, and ultimately in search of the greatest number of 'win-win' situations as possible, where the environment is also a key stakeholder"), is essential. We the question how critical social and environmental science approaches can have a space in engineering curricula, as these topics (particularly the social sciences), despite their importance, are traditionally shunned in engineering education.

## 7. Limitations of This Study

There are two major limitations in this study that could be addressed in future research. The first limitation is related to the number and selection of participants included in our interviews and workshops, which were relevant in terms of the qualitative analysis of the C&DW and e-waste contexts in Colombia and Argentina presented but not statistically representative for a quantitative analysis. The second limitation is related to the reduced scope of our curricular review, since courses in departments other than Mining Engineering should be explored, including Chemical Engineering, Environmental Engineering, Resources Engineering, and Materials Engineering, among others.

## 8. Conclusions

As illustrated above, urban mining has a leading role in the circular economy that is currently developing. However, in particular contexts, such as in Latin American countries, this activity poses additional benefits that can be maximized if they are understood and studied. We state that the study of urban mining from an interdisciplinary approach could contribute to this field in order to achieve a much more inclusive and sustainable activity.

For urban mining, cities are the locations where the extraction, circulation, and accumulation of materials take place. Hence, to favor inclusive urban mining it is not only necessary to understand collection and extraction processes but also to understand cities

and their context. Contextualizing this activity means analyzing local legal frameworks, stakeholders involved, their history, ideologies, culture, alliances and power differentials, the flow of materials, current technologies, and processes. In light of our findings, we argue that the future challenges associated with inclusive urban mining are sociotechnical in nature. Thus, we highlight the importance of promoting community-based research methods and concepts from the Engineering and Sustainable Community Development practices [65] to be included in mining, materials, metallurgical science, and engineering academic programs as a way to address these challenges.

It is not without reason that we have argued the case for working alongside communities to solve problems in a participatory way, as the knowledge of the local community members is crucial for developing sustainable and just solutions. However, this effort makes it necessary to promote knowledge sharing throughout the entire problem-solving process within and between multiple fields, disciplines, and communities and it also exemplifies the importance of fostering sustainable networking pathways. Productive interactions between groups are fundamental to maximizing the capacity to collaboratively find a viable, just, and long-term solution to community problems. To understand effective knowledge sharing, we argue that studying groups that are already doing this successfully is essential. Uniminuto, a Colombian University, is a prime example of an academic institution striving to create a positive social impact and uplift the vulnerable communities. Through knowledge and experience gained in PCIS projects, they have developed the "Social Innovation Route Framework" [40], a five-stage framework outlining community engagement projects, which is a powerful tool that academic institutions, especially those related to engineering and design, can utilize to take a proactive role in (1) working with vulnerable groups to improve their labor and environmental conditions and (2) understanding the sociotechnical dimensions of their projects.

Our observations also reflect the benefits of educational proposals that combine engineering knowledge with concepts from the sustainable community development framework, which is based on the social sciences. Thus, the interdisciplinary approaches that motivate students to make a contextual analysis of their projects, including history, politics, ideology, ethics, and culture, influence the way in which they develop their inventions. These approaches could also bring them closer to the Latin American context without falling into methodologies of the North–South dominance.

To answer the questions that were raised in this work, we propose some additional areas for future research. First, there is a need for a deeper analysis of the US and Latin American science engineering curricula to understand the lack of urban mining content and identify synergistic opportunities with overseas academic entities. Second, further work needs to be undertaken to screen current educational programs in the US since our study is limited. Third, future efforts within the engineering education field should be focused on developing an outline of an Inclusive Urban Mining course with the insights from this research that could be then included in the US engineering programs. Fourth, additional topics that should supplement the study of inclusive urban mining should be identified. Some topics that we believe could be beneficial to include are understanding the material politics of what is traditionally viewed as "waste" as well as learning social science approaches, such as contextual and empathetic stakeholder engagement strategies, to properly understand cities and their context.

**Author Contributions:** Conceptualization, S.L.S. and J.E.S.; Data curation, S.L.S. and J.E.S.; Formal analysis, S.L.S. and J.E.S.; Funding acquisition, S.L.S. and J.E.S.; Investigation, S.L.S. and J.E.S.; Methodology, S.L.S. and J.E.S.; Project administration, S.L.S. and J.E.S.; Resources, S.L.S. and J.E.S.; Validation, S.L.S. and J.E.S.; Writing—original draft, S.L.S. and J.E.S.; Writing—review and editing, S.L.S. and J.E.S. All authors have read and agreed to the published version of the manuscript.

**Funding:** The studies were funded by the Colorado School of Mines (Mines) Humanitarian Engineering and Science Program (Humanitarian Engineering & Science Grant 2022).

**Institutional Review Board Statement:** Both projects were approved by the Colorado School of Mines Human Subjects Research Team and exempted from the Institutional Review Board (IRB) process requirements.

**Informed Consent Statement:** Informed consent was obtained from all participants involved in both studies.

**Data Availability Statement:** Some data presented in this study are available on request from the corresponding authors. These data are not publicly available to protect the identity of participants. Other data presented in this study are available in the sources listed in References.

**Acknowledgments:** The authors would like to acknowledge and express their gratitude to the Colorado School of Mines (Mines) Humanitarian Engineering and Science Program for their support, the Corporación Universitaria Minuto de Dios (Uniminuto) teams that supported the Colombia project, the Fulbright Commission and the Ministry of Education in Argentina that supported one of the researchers' education, and the members of e-waste programs and cooperatives that contributed with their time, patience, and knowledge. Any opinions, findings, and conclusions or recommendations expressed in this material are those of the author(s) and do not necessarily reflect the views of the supporting organizations. The authors would also like to acknowledge Juan Lucena (Mines) for his tireless work, advising the authors and continuously presenting us with opportunities for our personal and professional development and Martha Liliana Herrera Gutiérrez (Uniminuto), who was in charge of translation, facilitation, and communication in the Colombia project. We express our gratitude to the community members in Argentina and Colombia and the interviewees that made this research possible.

**Conflicts of Interest:** The authors declare that they have no competing financial interest or personal relationship that could have appeared to influence any part of their work as reported in the paper. The funders had no role in the design of the study; in the collection, analyses, or interpretation of data; in the writing of the manuscript; or in the decision to publish the results.

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
