# Peer review of "Inclusive Urban Mining: An Opportunity for Engineering Education"

_mining, doi:10.3390/mining3020018_

Round 1
Reviewer 1 Report
1. In the Introduction, the term “electrical and electronic waste” must be changed to the “waste electrical and electronic equipment (WEEE. or called as e-waste)”.
2. To highlight the motivation of this study, the authors may provide the brief description about the status of WEEE Directive in EU.
3. In the Sec Objectives, the novelty of this study should be highlighted according to the survey on less/few works by other authors.
4. In the Sec. 3.1. and the Sec. 3.2, the authors should summarize the status (e.g., generation amount, recycling or reuse ratio) of C&DW in Columbia and e-waste in Argentina, respectively.
5. In the Sec Materials and Methods, the authors should provide a flowsheet for summarizing the logical items of this study
6. In the Sec. Inclusive urban mining in the US engineering curricula, the authors only searched the curricular databases from universities in the United States with Mining Engineering Bachelor of Science programs. However, this course may be assigned to other departments, including Chemical Eng., Environmental Eng., Resources Eng., Chem. & Environ. Eng., Environ. Resource Management, and so on.
7. The authors may provide the outline of the course Inclusive Urban Mining.
Author Response
- In the Introduction, the term “electrical and electronic waste” must be changed to the “waste electrical and electronic equipment (WEEE. or called as e-waste)”.
Response:
- Changed in Abstract (line 15). It read “e-waste” and now: “WEEE (or e-waste)”
- Changed in Introduction (lines 43 and 44). It read: “electrical and electronic waste (e-waste)” and now it reads “waste of electrical and electronic equipment (widely known as WEEE or e-waste)”.
- To highlight the motivation of this study, the authors may provide the brief description about the status of WEEE Directive in EU.
Response:
- Added in lines 65-72. The text now reads: “Although it is promising as an economic, social, and environmental activity, urban mining still has its limitations. The recent efforts in legislation to promote urban mining that have been implemented in Europe and other regions –for example, the WEEE Directive (2012/19/EU) and its recent amendments that have become international models for e-waste management [10–12]– are not enough to deal with the 82% of electrical and electronic equipment that is not treated in a sound environmentally manner [7,12] and the 35% of C&DW that still ends up in landfills globally [13].”
- In the Sec Objectives, the novelty of this study should be highlighted according to the survey on less/few works by other authors.
Response:
- The following paragraph has been added in lines 148-151: “Given the few works found in the literature that introduce these ideas and perspec-tives, we challenge the status quo by proposing a conversation on a new paradigm that completely changes how we understand and treat material stocks. In this regard, we un-derstand that the novelty of this study should be highlighted.”
- In the Sec. 3.1. and the Sec. 3.2, the authors should summarize the status (e.g., generation amount, recycling or reuse ratio) of C&DW in Columbia and e-waste in Argentina, respectively.
Response:
- Section 3.1:The following was added to lines 158-164: “An estimated 35% of C&DW is disposed of in landfills without further treatment [32]. In their article, Colorado et al. attempt to obtain the first quantified values of C&DW in Colombia [13]. However, information on the management of C&DW in Colombia is very scarce, and Colorado et al. concluded that no reliable data depicting the amount of C&DW generated annually in Colombia exists. Similarly, most countries in Latin America do not collect data on the generation and quantification of C&DW [13].”
- Section 3.2: The following was added in lines 196-198: “Despite the fact that Argentina had the Latin American highest generation of e-waste in 2019 (328 kt) of the 13 countries studied by Wagner et al. [35], the management of this waste stream is considered an emergent activity in the country, and little is known about it.”
Furthermore in Section 5.2.1 (lines 578-580) we included the following: “Regarding e-waste in Argentina, a national report estimates that 465,000 tons of this waste stream are generated per year [34] and only 4% is managed in an environmentally sound manner [35].”
- In the Sec Materials and Methods, the authors should provide a flowsheet for summarizing the logical items of this study
Response:
- A flowsheet was included in Section 4. Methods, below line 244.
- In the Sec. Inclusive urban mining in the US engineering curricula, the authors only searched the curricular databases from universities in the United States with Mining Engineering Bachelor of Science programs. However, this course may be assigned to other departments, including Chemical Eng., Environmental Eng., Resources Eng., Chem. & Environ. Eng., Environ. Resource Management, and so on.
Response:
- Because of time constraints, the research in other curricular databases was not conducted. However, the following paragraph was included in a new section “ Limitations of this study” (lines 784-792): “There are two major limitations in this study that could be addressed in future research. The first limitation is related to the number and selection of participants included in our interviews and workshops, which were relevant in terms of the qualitative analysis of the C&DW and e-waste contexts in Colombia and Argentina presented but not statistically representative for a quantitative analysis. The second limitation is related to the reduced scope of our curricular review, since courses in other departments different than Mining Engineering should be explored, including Chemical Engineering, Environmental Engineering, Resources Engineering, and Materials Engineering, among others.”
- The authors may provide the outline of the course Inclusive Urban Mining.
Response:
- We consider that it is out of the scope of our study to recommend an outline; however, in section 8. Conclusions, we included the following text (lines 836-844): “To answer the questions that were raised by this work, we foresee some additional areas for future research. (...) Third, future efforts within the engineering education field should be focused on developing an outline of an Inclusive Urban Mining course with the insights from this research that could be then included in US engineering programs."
---------------------------------------------------------
ADDITIONAL GENERAL CHANGES:
- The grammar, spelling, clarity, punctuation, and conciseness have been reviewed again.
- Titles changed:
- Materials and methods → Methods (line 235)
- Participatory framework and ethical considerations → Methodological framework and ethical considerations (line 236)
- Additional information added:
- “and lower the mining footprint” (line 54)
- “small and medium-sized enterprises (SMEs)” (80)
- “cities play a role in offering stable, secure, formal, economically sufficient, and dignified green jobs” (132)
- “urban miners are or can be empowered to build livelihoods” (139)
- Additional information was added about the workshop on C&DW in Colombia in section 4.4 lines 333-340
- “the Minamata Convention forced countries to focus on reducing and, when feasible, eliminating the use of mercury” (lines 729-730)
Reviewer 2 Report
This is a good ms. Topical issues addressed,brings together a very wide range of relevant ideas, text is clear, sequencing and section titles clear. Some minor points to consider:
Need to standardise references in bibliography, information for books is very limited eg publisher, location where the book can be accessed.
Number of interviewees (4.2) small, not entirely clear how they were selected. This is mentioned towards the end of the ms: maybe better to have a Limitation section?
There is more detail than is needed for an international readership eg 'Peggy'. Less personal information for the big number of people you mention would tighten up your ms.
466 burning plastic waste creates major health problems for those doing it and others, should be mentioned in your text
5.2.1 Why not use C&DW all the time, you spell out 'waste' for no obvious reason
Author Response
This is a good ms. Topical issues addressed,brings together a very wide range of relevant ideas, text is clear, sequencing and section titles clear.
Thank you very much, we appreciate your time and valuable feedback.
Some minor points to consider:
1) Need to standardise references in bibliography, information for books is very limited eg publisher, location where the book can be accessed.
Response:
- We have used the software Zotero and followed the MDPI style for our references (quick rules here https://www.mdpi.com/authors/references and guidance here https://mdpi-res.com/data/mdpi_references_guide_v5.pdf). All the references have been re-checked, changed, and complemented accordingly.
2) Number of interviewees (4.2) small, not entirely clear how they were selected. This is mentioned towards the end of the ms: maybe better to have a Limitation section?
Response:
- The following paragraph was included in a new section “ Limitations of this study” (lines 784-792): “There are two major limitations in this study that could be addressed in future research. The first limitation is related to the number and selection of participants included in our interviews and workshops, which were relevant in terms of the qualitative analysis of the C&DW and e-waste contexts in Colombia and Argentina presented but not statistically representative for a quantitative analysis. The second limitation is related to the reduced scope of our curricular review, since courses in other departments different than Mining Engineering should be explored, including Chemical Engineering, Environmental Engineering, Resources Engineering, and Materials Engineering, among others.
3) There is more detail than is needed for an international readership eg 'Peggy'. Less personal information for the big number of people you mention would tighten up your ms.
Response:
- The following paragraph in section 5.1.1., lines 429-439, was deleted “One morning in mid-June, we had the privilege to speak with Peggy. Peggy is one of the natural leaders of Community A. We were lucky enough to work with Peggy on this project and spent a lot of time on her porch, doing interviews with community members, meeting and talking with her family, or observing how the community naturally operated. We talked about her family, how she came to Community A, what she thought about the project, and what she was hoping for, especially for the women in the community. When asked about women’s empowerment, she stated, “[Women’s empowerment] is the idea that women can work on their own with their own capacities.” Bringing up examples such as doing their own nails or making their own clothes. Peggy brought up the importance of bringing opportunities to the community and doing workshops to give women tools to find jobs.”
- The quote “When asked about women’s empowerment, one leader in the community stated, “[Women’s empowerment] is the idea that women can work on their own with their own capacities.” She also discussed the importance of bringing opportunities to the community and doing workshops to give women tools to find jobs.” was added to the end of the first paragraph in section 5.1.1, lines 424-428.
4) 466 burning plastic waste creates major health problems for those doing it and others, should be mentioned in your text.
Response:
- The following paragraph has been modified in lines 508-509: “Others have perfected techniques, such as manual disassembly or burning, to improve the quality of the metals they obtain, even in conditions that create major health problems for them and their communities because of their exposure to hazardous chemicals [44].
5) 2.1 Why not use C&DW all the time, you spell out 'waste' for no obvious reason
Response:
- Changes in lines 307, 309, 522, 525, 528, 531, 532.
_______________________________________________________________________________
ADDITIONAL GENERAL CHANGES:
- The grammar, spelling, clarity, punctuation, and conciseness have been reviewed again.
- Titles changed:
- Materials and methods → Methods (line 235)
- Participatory framework and ethical considerations → Methodological framework and ethical considerations (line 236)
- Additional information added:
-
- “and lower the mining footprint” (line 54)
- “small and medium-sized enterprises (SMEs)” (80)
- “cities play a role in offering stable, secure, formal, economically sufficient, and dignified green jobs” (132)
- “urban miners are or can be empowered to build livelihoods” (139)
- Additional information was added about the workshop on C&DW in Colombia in section 4.4 lines 333-340
- “the Minamata Convention forced countries to focus on reducing and, when feasible, eliminating the use of mercury” (lines 729-730)
-
Round 2
Reviewer 1 Report
Based on the author's replies, it can be accepted in current form.